# U-NET BASED INDOOR RADIO MAP PREDICTION UNDER SPARSE SAMPLING

*Tianxiang Xing*[*]     *Leyi Zou*[*]     *Tejas Bharadwaj*     *Rushabha Balaji*     *Danijela Čabrić* [†]

Electrical and Computer Engineering Department, University of California, Los Angeles, USA

## ABSTRACT

In this paper, we present a runtime-efficient method for 2D pathloss (PL) map prediction in complex indoor environments, based on the U-Net convolutional neural network. The proposed method reconstructs full PL maps assisted by sparse measurements and preprocessed environment-aware geometrical features in highly-cluttered environments. We empirically show that such features help the network not only generalize to unseen points in the same environment but to different environments as well. Some of these features include the obstruction count map, accumulated transmittance maps, free-space pathloss and log-scaled distance map, which are collectively used as input features to the network. Our method is evaluated in the context of MLSP 2025 The Sampling-Assisted Pathloss Radio Map Prediction Data Competition. The evaluation results demonstrate that the proposed method achieves a weighted final root mean square error of 4.80 dB with an average total runtime of 14.36 milliseconds.

***Index Terms***— Radio map prediction, U-Net, sparse sampling, pathloss

## 1. INTRODUCTION

Accurate prediction of pathloss (PL) maps in indoor wireless environments is fundamental to a wide range of applications including user-cell site association, fingerprint-based localization, and path planning [1]. Traditionally, indoor PL maps are generated using deterministic ray tracing simulators, which model electromagnetic wave propagation using known environment maps [2, 3]. Such ray tracing simulators provide highly accurate results, but are computationally expensive and time-consuming. Although methods such as Instant-RM [4] can be very fast, like traditional ray-tracing software they require a detailed 3D-map of the environment. Though the cost of the 3D map generation can be amortized for static environments, due to the dynamic nature of indoor environments, these traditional methods can become impractical for use. Additionally, disparity between simulated and real-world measurements cannot be used to inform and improve the simulators. To effectively and efficiently incorporate real-world measurements, data-driven methods are widely adopted as practical alternatives to simulation-based approaches [5]. More importantly, for these data-driven methods to replace the traditional methods, they need to generalize to new points in the same environment and preferably to different environments as well. As a first step in achieving such environment-agnostic models, the PL prediction problem in 3D is relaxed to that of its 2D counterpart. 2D maps are more readily available than the 3D maps, and are easier to update as well. Learning-based models, particularly convolutional neural networks (CNNs), can use 2D maps along with the sparse measurements to learn a mapping from the environment to its corresponding PL map. Such neural network architectures have been shown suitable for PL prediction but with additional environmental priors [6].

In this work[1], we use 2D environmental priors along with sparse measurements to predict the 2D PL map of the indoor environment. We use the dataset and the corresponding benchmarking constraints in [7] to evaluate our learning-based PL reconstruction methods. The dataset consists of 2D environmental priors including reflectance and transmittance parameters of the different objects in the environment, which are encoded as images [8]. Our model is evaluated on the following two tasks, *Task 1:* Evaluate the reconstruction performance of our model under a fixed sampling methodology. For each distinct map layout, a unique set of measurement locations is randomly generated. *Task 2:* Evaluation of model under custom sampling methodology. Both tasks are performed for each of the two different sampling densities, 0.5% and 0.02%. We develop a lightweight U-Net-based model that incorporates multiple domain-informed priors including obstruction count map (OC map), accumulated transmittance maps ($\mathbf{T}_{sum}$), free-space PL (FSPL), and log-scaled distance along with sparse PL measurements. Our model achieves high reconstruction accuracy and fast inference speed, making it suitable for practical deployment in indoor RF systems.

---

[*]Authors Contributed Equally
[†]Corresponding Author

[1]This work was part of the MLSP 2025 The Sampling-Assisted Pathloss Radio Map Prediction Data Competition

## 2. DATASET AND FEATURE ENGINEERING

### 2.1. Dataset

For reader's reference, we give a brief exposition of the dataset, while more details can be obtained in [8]. The training dataset consists of PL maps corresponding to three different frequencies, namely 868 MHz, 1.8 GHz, and 3.5 GHz. The training set includes 3750 PL maps generated across 25 unique indoor layouts, each with 50 unique transmitter positions. The testing dataset contains PL maps corresponding to only 868 MHz. The test set comprises 200 PL maps from 5 unique layouts, different than the training dataset, with 50 transmitter positions for 3 layouts and 25 transmitter positions for the other 2 layouts. The ground truth (GT) PL map is a 2D gray-scale whose pixel values represent the signal attenuation in decibels (dBs). Along with the GT PL map (label), the dataset consists of the following 2D environmental priors (i) reflectance map of the objects, (ii) transmittance map of the objects. Additionally, the location of the transmitter in the environment is used as prior information to regress the PL values. All the above maps have the same resolution, and the spatial resolution is fixed to 0.25 m per pixel. Note that the image dimensions vary depending on the physical size of each indoor environment. A key characteristic of the dataset is that along with the environmental priors, PL measurements, sampled at either 0.5% or 0.02% of the total pixels is made available. In Task 1, these sampling points are selected randomly. In Task 2, a custom sampling strategy is used to obtain the sparse measurements.

### 2.2. Data preprocessing

A key insight behind the following features is to align the network's inductive bias with the inherent radial symmetry observed in the attenuation patterns of PL maps. In this regard, our feature engineering is informed with the geometrical structure of rays. A fundamental underpinning is to sketch rays from the transmitter to each pixel where the PL is to be predicted. Given the discrete nature of the inputs, the ray has to be discretized and in the next section we outline an efficient and GPU accelerated algorithm to do this.

#### 2.2.1. Accelerated Ray Sketching

We adopt an accelerated version of the classic 2D Bresenham's lines algorithm [9]. The Bresenham algorithm computes the sequence of integer grid cells that approximate a straight line between two points, using only integer arithmetic and comparison operations. To enable large-scale parallel processing, we implement a GPU-accelerated batch version of the Bresenham algorithm using CuPy [10]. For a given indoor scene, we simultaneously compute all paths from the transmitter to every pixel on the grid. This fast and accu-

rate Bresenham algorithm implementation serves as a foundational step for further data pre-processing methods.

#### 2.2.2. Obstruction count map

In PL prediction tasks, line of sight (LOS) masks are widely used to indicate whether a direct path exists between the transmitter and each receiver location [11]. While effective in distinguishing fully visible from fully obstructed regions, such binary representations fail to capture the degree of obstruction, treating all non-LOS pixels equally regardless of how many objects the rays must traverse. To better exploit the spatial information embedded in the environment layout, we introduce the obstruction count (OC) map [12]. Each pixel value is the count of object intersections encountered along the direct path of a ray originating from the transmitter to that pixel. This representation allows the model to distinguish between lightly and heavily obstructed paths, offering a more fine-grained prior about signal power degradation. An example of an OC map is shown in Fig. 1. The OC map serves as one of the feature maps to our model.

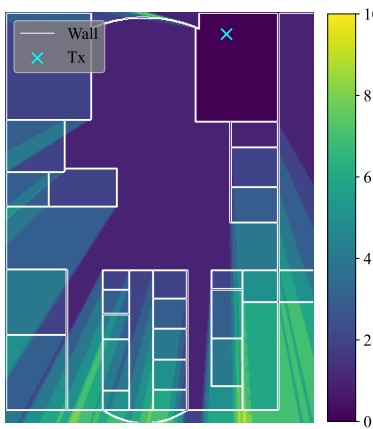

**Fig. 1**. Example of an OC map. The number indicates the exact OC value at each pixel.

#### 2.2.3. Transmittance Feature Map

While the OC map provides geometric information, it implicitly assumes all objects have the same attenuation properties. In practice, however, different materials exhibit varying degrees of transmittance attenuation. To account for this, we introduce the transmittance sum ($\mathbf{T}_{sum}$) map based on Bresenham algorithm, which aggregates the transmittance value (in dB) of each object encountered by a ray from the transmitter to a given pixel [13]. This sum reflects the cumulative attenuation experienced by the ray solely based on the physical materials encountered along its propagation trajectory. An example of a $\mathbf{T}_{sum}$ map is illustrated in Fig. 2. The distance based signal attenuation is not captured by $\mathbf{T}_{sum}$ map, to in-

corporate this attenuation we use the FSPL. The FSPL is modeled using Friis equation [14], and in the dB scale is given as

$$\text{FSPL}(d, f) = 20 \log_{10} \left( \frac{4\pi f d}{c} + \epsilon \right), \tag{1}$$

where[2] $d$ is the Euclidean distance (in meters) between the transmitter and the receiver, and $f$ is the frequency, with $c = 3 \times 10^8$ m/s. We perform an element-wise addition of the $\mathbf{T}_{\text{sum}}$ map and the FSPL map (both in dB units) to incorporate distance-based attenuation. Hereafter, $\mathbf{T}'_{\text{sum}}$ map refers to the combined attenuation of $\mathbf{T}_{\text{sum}}$ and FSPL values.

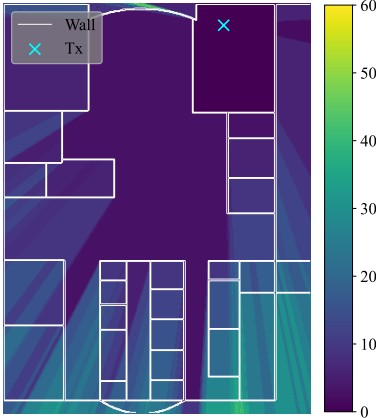

**Fig. 2**. Example of a $\mathbf{T}_{\text{sum}}$ map (dB). Each value represents the cumulative transmittance along the path from the transmitter to the corresponding pixel.

*Remark:* The introduction of the $\mathbf{T}'_{\text{sum}}$ map serves two key purposes. First, it incorporates how each ray is attenuated as it intersects and propagates through objects along its path. Second, compared to the original raw transmittance map—which is static and independent of the transmitter location—the $\mathbf{T}'_{\text{sum}}$ map reflects transmitter-aware attenuation patterns that vary across spatial positions. This contrasts with the reflectance map, which provides a location-independent structural prior related to surface materials and boundaries. This path-integrated, directional feature introduces non-local geometric dependencies into the learning process, which allows the network to process propagation effects that depend on the transmitter-receiver path geometry, rather than on local appearance alone.

### 2.2.4. Logarithmic Distance Channel

The logarithmic distance channel is obtained for each pixel at a radial distance $d$, whose value is given by

$$d_{\text{norm}} = \log_{10}(d + \epsilon). \tag{2}$$

This logarithmic mapping encodes the distance based attenuation and provides signal attenuation independent of the frequency. More details of its utility is empirically demonstrated in section 4.

## 3. DEEP LEARNING MODEL

To recover dense PL maps from sparse measurements, we adopt a convolutional encoder–decoder architecture based on the U-Net architecture [15], which is well-suited for spatially structured image-to-image regression problems. To capture local context, average pooling is employed within the network. Additionally, dropout with a rate of 0.2 is applied at the bottleneck to enhance generalization and reduce overfitting. The input to the network is a five-channel tensor, which includes reflectance, the $\mathbf{T}'_{\text{sum}}$ channel, a log-scaled distance map, an OC map, and sparse PL measurements. The output is a single-channel map representing the predicted PL in dB. In the following subsections, we describe the loss function, training methodology, and fine-tuning strategy of our model.

### 3.1. Loss Function

The loss function used to train the model is a weighted RMSE loss. This loss is computed only on the pixels where the PL measurements are unknown. Let $H \times W$ denote the size of an indoor layout, $\hat{y} \in \mathbb{R}^{H \times W}$ denote the predicted PL map, $y \in \mathbb{R}^{H \times W}$ denote the GT PL map, and $M \in \{0, 1\}^{H \times W}$ be a binary mask indicating where $M_{i,j} = 1$ if the corresponding pixel $(i, j)$ is not sampled. The loss function is defined as

$$\mathcal{L}_{\text{RMSE}} = \frac{1}{||M||_F} \sqrt{\text{Tr}\left(M^T (y - \hat{y})^2\right)}, \tag{3}$$

where $\text{Tr}(.)$ denotes the trace operation and the Frobenius norm of a matrix $X$ is denoted by $||X||_F$.

### 3.2. Experiment Settings and Methodology

In our experiments, 80% of the dataset is selected for training and the remaining 20% is reserved for validation. The resulting split yields 3000 samples for training and 750 samples for validation.

#### 3.2.1. Input Normalization

Each input channel is individually normalized to ensure that the dynamic range is consistent across different features. This ensures stable training. The reflectance channel is scaled by a factor of 0.05. The $\mathbf{T}'_{\text{sum}}$ map is min–max normalized to the [0,1] range. The log-transformed Euclidean distance channel is divided by 2.5 to compress its scale. The sparse PL measurements are normalized by dividing by 100. Finally, the OC map is scaled by its maximum value to produce a smooth auxiliary input channel.

---

[2]The additive constant $\epsilon$ inside the logarithm prevents overflow when $d \approx 0$, near the transmitter, and compresses the dynamic range for more stable learning. We consider $\epsilon = 1$

### 3.2.2. Padding Strategy

Samples within a training batch must share the same dimensions in order to be collated into a single tensor. To accommodate this, we apply an additional dynamic padding step at the batch level. During training, each minibatch is padded to match the maximum height and width among its samples. All 5 input channels are padded with the value zero. To prevent the contribution of the padded regions to the loss value, we apply a binary mask. This mask is set to one for the padded pixels and for the pixels where ground-truth PL values are available (i.e., sampled points), and to zero elsewhere. It is important to note that the sampling mask is not an input channel, but a persistent variable maintained for calculating the RMSE. By applying this mask, supervision is restricted to unsampled regions within the original image boundary. The predicted output is cropped to recover the original spatial resolution.

### 3.2.3. Fine-tuning

The training dataset spans across all three frequency bands, while the test set contains only samples from the 868 MHz ($f_1$) band. To leverage this prior information, we adopt a two-stage training strategy. After training on the full training set covering all three frequency bands, we fine-tuned the model using only $f_1$ training samples with a smaller learning rate. This fine-tuning step adapts the model more precisely to the distribution of the target frequency band.

### 3.2.4. Training settings and hyperparameters

All experiments were conducted using PyTorch on a NVIDIA A100 Tensor Core GPU. Adam optimizer is used for training. The complete training configuration and hyperparameter settings are summarized in Table 1.

## 4. RESULTS

### 4.1. Task 1: Random Sampling

In Task 1, we evaluate the model under random sampling at two predefined rates: 0.5% and 0.02%. The training and validation RMSEs are reported before fine-tuning, while the test RMSEs are obtained after fine-tuning the model on the $f_1$ subset. A summary of the results is provided in Table 2, where the column headers indicate the stage of evaluation (Pre-Finetune vs. Finetuned).

To further illustrate prediction quality, Fig. 3 presents a visual comparison between the predicted and ground-truth PL maps in dB scale when the sampling density is 0.5%.

These results demonstrate that our method reconstructs the PL map with high fidelity under both sampling rates. Interestingly, we observe from Fig. 3 that prediction errors tend to accumulate along the lines connecting the transmitter (Tx)

**Table 1**. U-Net Architecture and Model Configuration

| Name | Symbol | Value |
|---|---|---|
| Input channels | $C_{\text{in}}$ | 5 |
| Output channels | $C_{\text{out}}$ | 1 |
| Encoder filter widths | — | $64 \to 128 \to 256$ |
| Decoder filter widths | — | $256 \to 128 \to 64$ |
| Bottleneck filter width | $W_{\text{b}}$ | 512 |
| Convolution kernel size | $k$ | $3 \times 3$ |
| Convolution padding | $d_{\text{pad}}$ | 1 |
| Dropout | $p$ | 0.2 |
| Pooling stride | $l_{\text{p}}$ | 2 |
| Upsampling stride | $l_{\text{q}}$ | 2 |
| Maximum epoch | $n$ | 100 |
| Batch size | $B$ | 4 |
| Learning rate | $\eta_0$ | $1 \times 10^{-4}$ |
| Learning rate for fine-tuning | $\eta_1$ | $1 \times 10^{-5}$ |

**Table 2**. RMSE (dB) on Train/Val/Test Sets for Task 1

| Sampling Rate | Pre-Finetune | | Finetuned |
|---|---|---|---|
| | Train | Validation | Test |
| 0.5% | 2.64 | 2.90 | 3.17 |
| 0.02% | 5.26 | 6.92 | 6.12 |

to corner regions in the layout. These corner points often exhibit sharp discontinuities due to diffraction and heavy obstruction, making these pixels difficult to predict. This error pattern provides a useful insight for Task 2.

### 4.2. Task 2: Adaptive Sampling

Our experiments reveal that uniform sampling outperforms random sampling. However, uniform sampling alone may overlook important structural features like corners, which tend to accumulate error due to sharp propagation discontinuities. To address this, for sample rate of 0.5%, we propose a hybrid sampling strategy that allocates 95% of the sampling budget to uniform points and 5% to corner points, extracted using the Harris detector [16] from the original reflectance channel. For sample rate of 0.2%, standalone uniform sampling is applied.

We evaluate this corner-aware sampling approach under the same two sparsity levels used in Task 1. The training and validation RMSEs are reported before fine-tuning, while the test RMSEs are obtained after fine-tuning on the $f_1$ subset. Table 3 summarizes all results, with column headers indicating the evaluation stage.

The results show that our corner-aware sampling strategy yields consistent improvements in training and validation RMSE compared to the fixed random sampling used in Task 1. This indicates that allocating sampling points to geo-

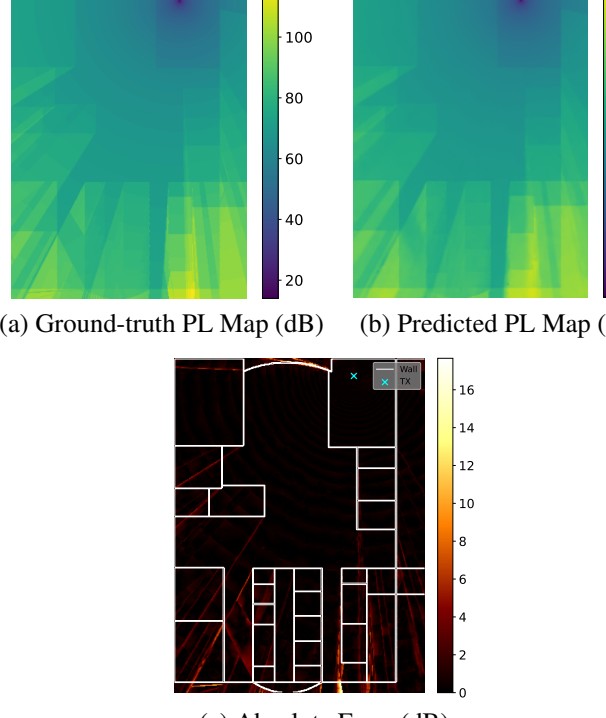

(a) Ground-truth PL Map (dB)    (b) Predicted PL Map (dB)

(c) Absolute Error (dB)

**Fig. 3**. Task 1 prediction quality of a scene under sampling rate of 0.5%.

**Table 3**. RMSE (dB) on Train/Val/Test Sets for Task 2 (Corner-Aware Sampling)

| Sampling Rate | Pre-Finetune | | Finetuned |
|---|---|---|---|
| | Train | Validation | Test |
| 0.5% | 2.42 | 2.54 | 3.22 |
| 0.02% | 4.47 | 5.57 | 6.84 |

metrically complex regions such as corners enhances learning within the observed data distribution. However, we observe a slight increase in test RMSE after fine-tuning, particularly at the 0.02% sampling rate. We attribute this to potential overfitting; the corner-aware strategy may induce sampling patterns that are overly tuned to the training layouts, thereby reducing the model's ability to generalize to unseen environments with different geometric structures.

### 4.3. Overall performance

In this section, we report the performance of our method in the MLSP 2025 Competition, where the average weighted RMSE is computed as follows [7]:

$$T = 0.3 \times (T_{1a} + T_{1b}) + 0.2 \times (T_{2a} + T_{2b}), \quad (4)$$

where $T$ denotes the average weighted RMSE, $T_{1a}$, $T_{1b}$, $T_{2a}$,

and $T_{2b}$ denote the test RMSE of Task 1 at sampling rates 0.02% and 0.5%, and Task 2 at sampling rates 0.02% and 0.5%, respectively. Our method achieves $T = 4.80$ and an average total runtime of 14.36 ms, of which 1.23 ms is used for model inference and 13.13 ms for data preprocessing.

### 4.4. Ablation Study

To assess the impact of various input features under the random sampling setting of Task 1 (with a 0.5% sampling rate), we conduct an ablation study using different feature combinations by selectively removing specific input channels. The configurations and corresponding results are summarized in Table 4. All models are trained using the same hyper-parameters as listed in Table 1.

From Table 4, we observe that systematically adding preprocessed features consistently improves model performance. In particular, incorporating hand-crafted environmental priors—such as physical features (e.g., log-distance, $\mathbf{T}'_{\text{sum}}$) and geometric structures (e.g., the OC map)—leads to steady performance improvement. Among all configurations, setting (v), which integrates the full set of engineered inputs (including modified transmittance, distance, and the OC map), achieves the lowest validation RMSE, which proves the necessity of $\mathbf{T}'_{\text{sum}}$ as mentioned in 2.2.3 and highlights the benefit of jointly using physical and geometric priors to enhance spatial generalization under sparse supervision.

Comparing configuration (v) with (vi), which excludes sparse PL samples, further emphasizes the value of sampled ground-truth observations. While prior-only models like (vi) can estimate coarse PL patterns, they fail to capture the fine-grained variations driven by layout-specific signal propagation. This confirms that even sparse PL inputs provide essential guidance for accurate map reconstruction. Overall, these findings validate our design strategy of combining domain-informed input features with limited but meaningful supervision to enable both accurate and generalizable indoor PL prediction.

## 5. CONCLUSIONS

This paper presents a sparse-supervision framework for pathloss (PL) map prediction, developed as part of MLSP 2025 The Sampling-Assisted Pathloss Radio Map Prediction Data Competition. Our method incorporates environment-specific priors which include physics-informed propagation effects—namely the $\mathbf{T}_{\text{sum}}$ map, free-space pathloss (FSPL), log-scaled distance map, and obstruction count (OC) map—into a lightweight U-Net architecture trained with sparse measurement samples at 0.5% and 0.02% sampling rates. Evaluation results demonstrate that the proposed method achieves a weighted average RMSE of 4.80 with an average runtime of 14.36 ms, highlighting its effectiveness under real-time constraints.

**Table 4**. U-Net Architecture and Model Configuration

| Configuration | Val RMSE (dB) |
|---|---|
| (i) Reflectance + Transmittance + Distance + Sampled GT | 3.2709 |
| (ii) Reflectance + Transmittance + Log-scaled Distance + Sampled GT | 3.2066 |
| (iii) Reflectance + Transmittance + Log-scaled Distance + OC Map + Sampled GT | 2.9645 |
| (iv) Reflectance + $\mathbf{T}'_{\text{sum}}$ + Log-scaled Distance + Sampled GT | 2.9320 |
| (v) **Reflectance + $\mathbf{T}'_{\text{sum}}$ + Log-scaled Distance + OC Map + Sampled GT** | **2.9004** |
| (vi) Reflectance + $\mathbf{T}_{\text{sum}}$ + FSPL + Log-scaled Distance + OC Map + Sampled GT | 2.9349 |
| (vii) Reflectance + $\mathbf{T}'_{\text{sum}}$ + Log-scaled Distance + OC Map | 15.1922 |

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
