# OpenReview forum: "U-Net Based Indoor Radio Map Prediction under Sparse Sampling"
_IEEE.org/MLSP/2025_SA_Radio_Map_Prediction_Challenge — SA Radio Map Prediction Challenge at MLSP 2025 Oral_

### Official Review · Reviewer_XnSK · 2025-06-05
**Review for ''U-Net Based Indoor Radio Map Prediction under Sparse Sampling''**

**Rating:** 8
**Confidence:** 4

**Review:**

This paper presents a runtime-efficient method for 2D path loss (PL) map prediction in complex indoor environments based on UNet. The proposed method reconstructs full PL maps from sparse measurements by incorporating well-designed environment-specific priors, such as the Obstruction Count Map and the Transmittance Sum Map, demonstrating a deep understanding of wireless propagation mechanisms. The method achieves superior performance in terms of reconstruction accuracy and runtime, indicating its potential for deployment.
Below are some revision suggestions for the authors' reference.

1. Throughout the paper, the terms "approach", "method", and "framework" are used interchangeably to refer to the proposed method. Please consider using consistent terminology.

2. To ensure uniform input sizes for different indoor layouts, the authors adopt dynamic padding: each mini-batch is padded to match the maximum height and width among its samples. Does this imply that the input sizes vary across batches? Have the authors compared this with an alternative approach where all inputs are resized or padded to a fixed size?

3. According to the results of the ablation study in Table 4, there is a significant performance gap between models with and without Sampled GT. Was the validation RMSE in the table also computed using Equation (3)? Using the original unmasked RMSE formula throughout would facilitate fairer comparisons.

4. How are the training and validation sets partitioned? Does the validation set contain data from the same layouts as in the training set, but with different transmitter positions?

5. In Section 2.1 Dataset, the final test set actually contains only 200 PL maps instead of 250. B1 and B5 have 25 samples each, while B2, B4, and B6 have 50 samples each.

6. There are some typographical issues. For example, in the abstract, the sentence "Some of these features include, the obstruction count map, accumulated transmittance maps..." contains an unnecessary comma after "include" and should be corrected. In addition, the terms "pathloss" and "path loss" are both used throughout the paper. Please ensure consistent usage.

---

### Official Review · Reviewer_ZsBD · 2025-06-05
**Review for ''U-Net Based Indoor Radio Map Prediction under Sparse Sampling''**

**Rating:** 8
**Confidence:** 4

**Review:**

This paper presents a well-designed U-Net-based approach for indoor radio map prediction, leveraging novel feature engineering such as the obstruction count map and transmittance feature map to achieve strong performance in both prediction accuracy and inference speed. The authors demonstrate impressive innovation by incorporating physics-inspired geometric features that align with signal propagation principles, resulting in a lightweight yet effective solution that achieves a weighted RMSE of 4.80 dB with remarkable computational efficiency. The comprehensive evaluation across multiple sampling densities and the systematic ablation study further validate the robustness of the proposed methodology. While the methodology and results are compelling, several points could be clarified or expanded to further strengthen the paper:

1. Fine-tuning Validation: The paper employs fine-tuning to adapt the model to the 868 MHz frequency band. However, the results in Tables 2 and 3 only compare pre-fine-tuning training/validation RMSE with post-fine-tuning test RMSE. Including a comparison of validation RMSE before and after fine-tuning would provide stronger evidence of its effectiveness, as it would demonstrate whether the fine-tuning process actually improves generalization on unseen data from the target frequency band.
2. Adaptive Sampling at Extremely Low Sampling Rates: The proposed adaptive sampling strategy allocates 5% of the sampling budget to corner points. At a 0.02% sampling rate, this could result in very few (or even zero) corner points being sampled. The paper does not address how the method handles such cases—for instance, whether it falls back to uniform sampling or employs another strategy. Clarifying this would strengthen the robustness of the approach.
3. Runtime Breakdown: The reported average runtime of 14 ms is impressive, but it would be helpful to separate the time spent on data preprocessing (e.g., ray sketching, feature map generation) from the actual model inference. This breakdown would provide insights into potential bottlenecks and highlight opportunities for further optimization.
4. Ablation Study Configuration :In configuration (vi) of the ablation study, the model is trained without sparse PL samples ("Sampled GT"). The paper states that the same hyperparameters from Table 1 are used, but it is unclear how the absence of PL samples is handled. If the "Sampled GT" input is simply an all-zero matrix, removing this channel entirely (and reducing the input dimensionality) might improve performance, as the zeroed inputs could introduce noise or mislead the model. Clarifying this design choice or testing the alternative approach would be valuable.

---

### Official Review · Reviewer_1afR · 2025-06-08
**Reviewer's comments on "U-Net Based Indoor Radio Map Prediction under Sparse Sampling"**

**Rating:** 7
**Confidence:** 4

**Review:**

The paper presents a clean and efficient U-Net-based architecture augmented with well-engineered environmental priors like the OC map, FSPL, Tsum, and log-distance. The model achieves excellent performance on the MLSP 2025 challenge dataset (4.80 dB weighted RMSE), with an inference time of 1.23 ms (plus 13 ms preprocessing), suitable for real-time applications.

The authors could consider the following comments for improving minor aspects of the paper:
- Even though specified in the text, include unit measure in Fig. 3 around colour bars.
- Reformat Table 1 for readability, especially by visually separating encoder and decoder configurations.

---

### Official Review · Reviewer_ksFF · 2025-06-09
**Review for ''U-Net Based Indoor Radio Map Prediction under Sparse Sampling''**

**Rating:** 7
**Confidence:** 4

**Review:**

1) "The proposed approach reconstructs full PL maps from sparse measurements" : The proposed method uses not only sparse measurements but also many other environment- and propagation model-related inputs.

2) "We add the T_sum map to the FSPL map to incorporate distance-based attenuation. Hereafter, T_sum map refers to the combined attenuation of Tsum and FSPL values." : The authors' intended meaning of adding the T_sum map to the FSPL map is unclear. Is it the literal, mathematical, element-wise addition of the two maps?

3) "Second, by transforming the original transmittance map into the Tsum map, we enable the network to disentangle reflectance and transmittance features more effectively": There is no evidence to support this statement. For example, what would happen if the authors used both the Tsum and FSPL maps as separate input features? Or how would the model perform when the initial T_sum (without combining with the FSPL map) was used instead?
It is also unclear how this approach could help the network "disentangle reflectance more effectively." There is no clear evidence of this, and the explanation is not sufficiently convincing.
Overall, it is possible that the improvements resulting from the transformation of the transmittance channel to the transmittance sum (T_sum) have not been correctly interpreted, and the sources of improvement have not been correctly identified.

4) Additionally, have the authors considered applying the sum approach to the reflectance channel and generating an R_sum map? Did the authors try this?